# Inequality in the Utilization of Breast Cancer Screening between Women with and without Disabilities in Taiwan: A Propensity-Score-Matched Nationwide Cohort Study

**DOI:** 10.3390/ijerph19095280

**Published:** 2022-04-26

**Authors:** Puchong Inchai, Wen-Chen Tsai, Li-Ting Chiu, Pei-Tseng Kung

**Affiliations:** 1Graduate Institute of Public Health, College of Public Health, China Medical University, Taichung 406040, Taiwan; u107050122@cmu.edu.tw; 2Department of Health Services Administration, College of Public Health, China Medical University, Taichung 406040, Taiwan; wtsai@mail.cmu.edu.tw (W.-C.T.); u9775851@cmu.edu.tw (L.-T.C.); 3Department of Healthcare Administration, Asia University, Taichung 41354, Taiwan; 4Department of Medical Research, China Medical University Hospital, China Medical University, Taichung 404332, Taiwan

**Keywords:** disability, breast cancer screening, inequality

## Abstract

Because of the difficulties in accessing medical care, the likelihood of receiving breast cancer screening may be low for women with disabilities. We aimed to investigate differences in the utilization of breast cancer screening among women with and without disabilities. Participants included women with and without disabilities from 2004 to 2010, and it was observed whether the participants had received a breast cancer screening during 2011 and 2012. Propensity-score matching was employed to match disabled women with non-disabled women (1:1). Data sources included the National Health Insurance Research Database, the Cancer Screening Database, and the Disability Registration File. Conditional logistic regression was performed to examine the odds ratios (ORs) that both groups would undergo breast cancer screening. The proportion of women with disabilities who received breast cancer screening was 18.33%, which was significantly lower than that of women without disabilities (25.52%) (*p* < 0.001). Women with dementia had the lowest probability of receiving a mammography examination (OR = 0.34; 95% CI: 0.28–0.43), followed by those with multiple disabilities (OR = 0.43; 95% CI: 0.40–0.47) and intellectual disabilities (OR = 0.45; 95% CI: 0.41–0.50). In conclusion, compared to women without disabilities, those with disabilities were less likely to undergo breast cancer screening.

## 1. Introduction

According to the study conducted by the Global Burden of Disease Cancer Collaboration, cancer is the second leading cause of death worldwide, with 9.56 million people suffering from cancer [1]. In addition, the World Cancer Research Fund/American Institute for Cancer Research reported that approximately 1.7 million new cases of breast cancer were reported worldwide, and breast cancer was ranked the fifth highest cause of death among women [2]. In Taiwan, the four most common cancers are lung, liver, colorectal, and breast cancer [3]. However, based on the latest cancer registry report in Taiwan, breast cancer is the most common cancer among Taiwanese women, with an age standardized incidence rate of 70.7 per 100,000 persons in 2014 [4].

In order to diagnose and treat breast cancer in its early stages, screening for breast cancer with mammography is needed. The concept of screening with mammography is to use X-ray imaging to detect breast cancer before a lump can be felt [5]. Although the survival rates for breast cancer vary worldwide, the rates have improved since the early detection of breast cancer and advanced medical care have been promoted in many countries [2]. In Taiwan, the Health Promotion Administration, Ministry of Health and Welfare, has promoted mammographic screening programs among Taiwanese female adults since 2002, and women at high risk were initially classified through questionnaires before they undergo a screening [6].

Nevertheless, a high probability of unsatisfactory health checks or medical care may occur among disabled people, owing to the difficulty in accessing healthcare services [7]. With reference to the study regarding inequalities in receiving breast cancer screening, the results indicated that mammography examinations need to be improved more among women with disabilities than in those without disabilities [8,9]. In 2016, there were 1.17 million people with disabilities, accounting for nearly 4.97% of the entire population of Taiwan. Among people with disabilities in Taiwan, about 507,399 people with disabilities were female, accounting for 43.36% [10]. This evidence reflects concerns regarding medical care utilization amid a growing number of individuals with disabilities, which may be the cause of the difference in breast cancer screening utilization between women with and without disabilities.

Based on previous research related to the inequities in the utilization of breast cancer screening among women with and without disabilities, the classification of disability types and severities, as well as the related factors, are limited for the analyses [11,12,13]. Consequently, the aim of this research was to analyze the difference in the utilization of breast cancer screening between women with and without disabilities in Taiwan. The key concept was designed to determine how to address the inequalities in the utilization of mammography examinations among women with different disability types and severities, as well as to determine other factors correlated with the use of mammography examinations. With the purpose of understanding the overall situation regarding the use of mammography examinations and to improve the health of women with disabilities in Taiwan, the aforementioned goal is needed to regulate an investigation at a national scale. Principally, the present study anticipates building on new evidence in the related fields by using the advantageous statistical technique to generate precise estimation.

## 2. Materials and Methods

### 2.1. Data Sources

Data sources included the National Health Insurance Research Database (NHIRD) from the Taiwan Ministry of Health and Welfare, the Cancer Screening Database from the Health Promotion Administration, and the National Disability Registration Database from the Ministry of the Interior. The National Health Insurance (NHI) program was initiated in 1995 to preserve the right to receive healthcare among Taiwanese, covering 99.9% of the entire population [14]. The NHI program covers most of the cost of medical expenses; however, beneficiaries are required to contribute a small amount of payment (called a copayment), as it is the way to reach high accessibility in regard to healthcare [15]. Free screenings are provided for four types of cancer, namely colorectal, oral, breast, and cervical cancers, as part of the national screening program every two to three years [16]. According to breast cancer screening, the Health Promotion Administration has subsidized free breast cancer screening for women aged 50–69 since 2004, and the screening program is conducted once every two years [17].

The Ministry of the Interior is responsible for maintaining the National Disability Registration Database. Regarding the scheme for evaluating different types and levels of disabilities that was conducted in hospitals, disability evaluation criteria were derived from the medical model that doctors identified potential disability beneficiaries from by assessing the body impairment in each individual [18]. According to Taiwan’s Disability Rights Protection Acts, the different types and severities were in medical terms based on 18 categories (including persistent vegetative state), as well as 4 levels of severity [19]. As stated in Article III of the Physically and Mentally Disabled Citizens Protection Act, the identification of people with disabilities is issued by the Department of Social Welfare to applicants with disabilities who visit designated hospitals for examination to determine their disability status to ensure that they are able to receive the related advantages [20].

### 2.2. Study Design and Study Population

A nationwide retrospective cohort study was conducted in the present study. The study population included women with disabilities (excluding persistent vegetative stage, PVS) and women without disabilities from 2004 to 2010, and it was observed whether the study participants underwent breast cancer screenings between 2011 and 2012. When analyzing the use of breast cancer screening, we excluded the study population who had breast cancer before 2011 by authorizing data on the Taiwan Cancer Registry. In particular, the propensity score matching method was used to balance the data in both groups [21]. According to the research purpose and age limit of breast cancer screening, propensity score matching was used to match women with and without disabilities (Figure 1). Propensity score matching was conducted using greedy nearest neighbor matching by digit without replacement to form the study participants with a 1:1 matching ratio on the propensity score in order to minimize the selection bias.

### 2.3. Variable Descriptions and Definitions

Seven variables were considered in the study. (a) Age groups: Participants were categorized into two groups, namely 50–59 and 60–69 years, respectively. (b) Disability types: There were 17 types of disability, namely moving functional limitation, visual impairment, hearing impairment, speech disorder, intellectual disability, multiple disabilities, dysfunction of primary organs, facial impairment, dementia, congenital disorders, chronic mental health conditions, balance disorder, intractable epilepsy, autism, chromosomal abnormalities, congenital metabolic disorders, and rare diseases. (c) Severity of disability: The severities were divided into four levels, namely mild, moderate, severe, and very severe. (d) Factors related to economic status: Monthly salaries were divided into six bands, with low-income households (defined by the government) listed as a reference group. (e) Factors related to environment: The urbanization level of residential areas was classified into seven bands (where Levels 1 and 7 represent the highest and lowest levels of urbanization, respectively) based on the three major components: (1) demographic characteristics (population density, ratio of population with college education or above, and ratio of population over 65 years old); (2) industrialization (agricultural population ratio); and (3) medical resources (the number of western doctors per 100,000 people) [22]. (f) Comorbid conditions: The Charlson comorbidity index (CCI) [23] was considered, along with medical claim databases, to minimize the potential confounding that may occur in the present study, and several health conditions were included and divided into four levels (0, 1, 2, ≥3) to represent the comorbid conditions in each participant. (g) Preventive health behaviors: Adults’ preventive care services can reflect the preventive health behaviors of each participant that are divided into two categories (yes and no). Regarding variables used for propensity score matching, the variables, including age, monthly salary, urbanization level, and CCI, were included for matching women with and without disabilities. About the relevant factors correlated to the use of mammography examination between women with and without disabilities, the following variables were included in the analysis: disability type, the severity of the disability, age, monthly salary, urbanization level, CCI, and adults’ preventive care service.

### 2.4. Statistical Analyses

Propensity score estimates were calculated by using a logistic regression model, which included individuals with disabilities and those without disabilities as dependent variables. We included age, monthly salary, urbanization level, and CCI in terms of independent variables. Additionally, characteristics among women with disabilities and those without disabilities were compared by using the chi-square test. The variables with p-values greater than 0.05 can be denoted by the test with similar characteristics. According to the descriptive analysis, the chi-square test was used to compare the differences in the utilization of breast cancer screening between women with and without disabilities. Furthermore, conditional logistic regression was performed to investigate the odds ratio (OR) that women with or without disabilities would undergo breast cancer screening. All statistical analyses were conducted by using SAS version 9.4 (SAS Institute, Cary, NC, USA).

## 3. Results

Regarding before matching, the results were presented in Appendix A. The propensity-score matching technique (in a 1:1 ratio) was used to create baseline characteristics between women with and without disabilities. A total of 248,230 participants were included after matching, and there was no statistically significant difference in age, monthly salary, urbanization level of residence area, and CCI score between the two groups (Table 1).

When comparing breast cancer screening utilization (Table 2), we found a lower proportion of women with disabilities receiving a mammography examination (18.33%) than those without disabilities (25.52%). Concerning disability type, compared with women without disabilities as a reference group, the lowest proportion of women receiving a mammography examination was found among women with dementia (6.88%), followed by multiple disabilities (10.48%), dysfunction of primary organs (13.39%), intellectual disability (13.40%), and balance disorder (15.84%). Furthermore, the findings showed that the proportion of patients receiving mammography decreased with the severity levels of their disability. Even if the percentage of people receiving a mammography examination decreased with age in both women with and without disabilities, women with disabilities aged 50–59 and 60–69 still presented a lower percentage of those receiving a mammography examination, which was 26.86% and 7.72%, respectively. The magnitude of people receiving a mammography examination decreased with a lower level of monthly salary, in which women with disabilities who had a low-income status showed the lowest magnitude of receiving a mammography examination (13.91%) compared to those without disabilities who had a low-income status (21.22%). Regarding the urbanization level of the residence area, the proportion of women with disabilities receiving a mammography examination was lower than that of women without disabilities, in which women with disabilities living in an area of Level 7 (the least urbanized) had a lower likelihood of receiving a mammography examination (17.68%) than women without disabilities (26.06%). In relation to the highest score of comorbid conditions, women with disabilities also exhibited the lowest proportion of receiving a mammography examination (9.63%) compared to women without disabilities (23.17%). With reference to people who received preventive care services for adults, women with disabilities had a lower proportion of receiving a mammography examination (29.16%) than those without disabilities (31.73%) (Table 2).

According to the conditional logistic regression analysis shown in Table 3, there were three adjusted models for our analyses. After adjusting for other variables (age, monthly salary, urbanization level, CCI score, and preventive care service), model A showed that women with disabilities were 0.72 times less likely to receive a mammography examination than those without disabilities. Based on model B, women with dementia had the lowest probability of receiving a mammography examination (OR = 0.34; 95% CI: 0.28–0.43), followed by multiple disabilities (OR = 0.43; 95% CI: 0.40–0.47), intellectual disabilities (OR = 0.45; 95% CI: 0.41–0.50), and dysfunction of primary organs (OR = 0.59; 95% CI: 0.56–0.62), compared to women without disabilities. Concerning the severity of disability, model C presented the lowest probability of receiving a mammography examination among women with very severe disability (OR = 0.39; 95% CI: 0.37–0.41), followed by severe (OR = 0.42; 95% CI: 0.40–0.45) and moderate disability (OR = 0.73; 95% CI: 0.71–0.76), compared to women without disabilities (Table 3).

## 4. Discussion

Based on the present study, the proportion of the use of mammography examinations among disabled people (18.33%) was particularly lower than among non-disabled people (25.52%) (*p* < 0.001). Our findings were consistent with those of previous studies, which revealed that the probability of breast cancer screening was possibly low for disabled people, due to the difficulty accessing medical care [24,25].

Regarding different types of disabilities in the present study, when compared with women without disabilities, lower percentages of the use of mammography examinations were found in the group of women with dementia (6.88%), multiple disabilities (10.48%), dysfunction of primary organs (13.39%), and other types of disability. Furthermore, our results revealed that women with dementia showed the lowest probability of receiving a mammography examination (OR = 0.34; 95% CI: 0.28–0.43) compared with those without disabilities; these findings are consistent with the results of a previous meta-analysis of breast cancer screening among women with cognitive impairment or dementia, in which women with cognitive impairment or dementia showed a lower rate of screening for breast cancer by mammography compared with those without disabilities (pooled OR = 0.81) [26]. With reference to the study related to the effect of cognitive impairment on breast cancer screening, only approximately 26% of women with dementia or severe cognitive impairment received mammography examinations [27], whereas the utilization of mammographic examinations among some may be discontinued, due to a lower life expectancy or facing greater impairment [28,29]. In addition, a multi-ethnic population-based study reported that the average survival time for people diagnosed with dementia was approximately 5.7 years [30]; consequently, those with dementia were more likely to have less longevity, which may affect their likelihood of obtaining healthcare services, including breast cancer screening. In fact, caregiver involvement in the screening process was important for women with dementia [28]; nevertheless, women with dementia can still communicate their refusal [31]. These issues may influence the difficulty in receiving breast cancer screening among women with dementia. Consequently, based on our findings and the abovementioned studies, it is possible that those suffering from dementia were unlikely to undergo mammography examinations at the indicated periods, and this effect may lead to a small proportion and probability of the use of mammography examinations among women with dementia. In the future, we hope that policymakers will list women with dementia as one of the marginalized populations that should be considered to improve the possibility of receiving breast cancer screenings by using mammography examinations. In particular, in order to follow the breast cancer early detection recommendations, women with dementia should be encouraged to undergo breast cancer screenings at the recommended time. Caregivers and female health specialists could play an important role in providing support in the examination process among those with dementia, as well as to encourage them to undergo the examination without nervousness.

According to a survey among disabled women between the ages of 40 and 79 in the US, those with disabilities were less likely to receive a physician recommendation for breast cancer screening by mammography examination; this was obviously also found among the elderly and those with multiple disabilities [11]. In addition, regarding the qualitative research conducted among American women with multiple sclerosis, 80% of those with mobility limitations did not receive breast cancer screenings, and some of the participants in this previous study reported that they had problems that correlated with traveling from their homes to breast cancer services, as well as being uncomfortable [32]. Furthermore, a study on cancer screenings among women with mobility disabilities also reported that barriers to receiving cancer screenings, including mammography examinations, were found in a group of participants due to several conditions, such as transportation problems and cancer education [33]. Multiple sclerosis refers to an illness that can impact the center of consciousness, as well as the nervous system, triggering several indications, including visual difficulties, mobility limitations, and problems of consciousness or stability [34]. Hence, with reference to the aforementioned evidence, women with multiple disabilities, including visual impairment, balance disorder, and intellectual disability, may have a lower likelihood of receiving mammography examinations. Consistently, our findings also presented the lower probability of receiving mammography examinations among women with multiple disabilities (OR = 0.43; CI: 0.40–0.47), intellectual disability (OR = 0.45; CI: 0.41–0.50), dysfunction of primary organs (OR = 0.59; CI: 0.56–0.62), and balance disorder (OR = 0.63; CI: 0.45–0.89). Since the mobile health services that were established by the Taiwanese government for people living in rural areas that cannot afford health services, accessing transportation to receive mammography examinations is not a major problem among Taiwanese [35]. Nevertheless, based on the recommendations from a previous study on cancer screening among women with mobility disabilities [33], in order to increase the probability of accessing breast cancer screening services in a group of women with multiple disabilities, including intellectual disability, dysfunction of primary organs, and balance disorder, breast cancer education for healthcare providers and recipients would be a resolution that the government should take into consideration to improve access to appropriate breast cancer education for both healthcare providers and recipients.

Some studies have found a correlation between disability levels and a low likelihood of breast cancer screening in women with disabilities and women without disabilities [9,36]. This is following our findings indicating that women with a high degree of disability have a decreased likelihood of receiving breast cancer screening. Consequently, in order to enhance the likelihood of accessing breast cancer screening services among women with different severities of disability in Taiwan, some recommendations related to health-policy-making from previous studies, such as certifying the availability of information, transportation facilities, and accessing the common source of healthcare in some areas, may be established to diminish the difference between women with and without disabilities [9].

Remarkably, when compared with women living in urban areas, those living in rural areas showed a higher likelihood of receiving breast cancer screenings. Based on previous research on dental care utilization, the researchers found that children residing in rural areas were more likely to utilize mobile healthcare services to access dental care [37]. Nevertheless, the accessibility of healthcare among disabled women is difficult even in urban areas; to make life easier for those with disabilities in the future, the government should pay attention to this issue.

As reported by the research conducted among the American population aged 51 years and older, the study found that older people who had more negative self-perceptions were more likely to delay receiving healthcare and to experience healthcare barriers after adjusting for predisposing enabling and need factors. Likewise, the reasons for this are a lack of access to healthcare, being too busy to visit a physician, and a dislike of visiting a physician [38]. Similar to our findings, the tendency of a barrier to healthcare was found among the elderly, in which older women aged 60–69 years showed a lower percentage and probability of receiving a mammography examination than those aged 50–59 years. In particular, older women aged 60–69 years predominantly presented the lowest probability (OR = 0.27; 95% CI: 0.26–0.28) of receiving a mammography examination.

Based on our findings, the factors that also influenced the utilization of breast cancer screening were the levels of comorbid conditions, in which women with higher levels of comorbid conditions presented a lower proportion and probability of receiving breast cancer screening by using mammography examination. According to the results of previous studies related to self-management among older adults with several comorbidities, which stated that by hampering self-management in one or more ways, barriers could be thought of as undesirable properties related to poor health conditions; lower perceived health status was also associated with higher morbidity, lower physical function, less medical knowledge, fewer social activities, and financial hardship. Additionally, a higher level of morbidity, financial constraints, communication difficulties between patients and physicians, and low income were significantly more likely to create barriers to self-management when physical functioning was lower [39]. As a result, the degree of encouraging self-perceptions of aging in preventive health promotion may influence the decision to seek care for troubling symptoms among older adults and those with comorbidities. In the future, promoting positive self-perceptions of aging could motivate older women and those with comorbidities to act more proactively when it comes to their healthcare requirements.

According to studies on behavior and preventive medicine, behavior is central to the development, prevention, treatment, and management of preventable manifestations of disease and health conditions [40]. Preventive health behavior (PHB), or the activities accepted by people who believe they are in good health intended to prevent illness [41], is one of the habits that influence how often people visit a physician [42]. Reasons for taking part in PHBs may be powerlessness owing to low socioeconomic status, including income [43], which was related to our findings that women from low-income households had the lowest percentage and likelihood of receiving mammography examinations; moreover, a lack of education [44], lack of consciousness, and considerate healthcare issues [45] were important issues that influenced PHBs. Predominantly, our findings showed that women who received adults’ preventive care services showed a higher proportion and probability of receiving mammography examinations than those who did not. This reflected that PHB had an effect on the likelihood of receiving mammography examinations since those who were concerned about their health were more likely to receive healthcare examinations to know their health status. Consequently, changing health-related behaviors is a noteworthy intervention that must be considered for future health policy advice. However, in order to reduce inequalities, interventions aimed at encouraging health behavior changes should be managed at multiple levels outside the health sector [46].

Around 20% of the women in the study received mammography examinations. This outcome is remarkably low compared with Western countries, where screening adherence rates are regularly over 50% in the same age group. [47]. Consequently, improving the utilization rate of breast cancer screening among Taiwanese women by creating health promotion campaigns will be crucial to reducing breast cancer in the future. Moreover, developments in computer-aided diagnosis and artificial intelligence for breast cancer screening might be the alternative methods [48] that can carry out in parallel with the traditional approach to increase the rate of breast cancer screening, as well as a chance to detect breast cancer early among women with and without disabilities in the future.

In summation, our study could serve as a reference for government and policymakers to provide appropriate preventive health education, improve health and well-being, and reduce inequality in healthcare utilization among populations, especially women with disabilities, in order to follow the Sustainable Development Goals, which make several references to disability [49]. With reference to the announcement from the WHO for persons with disabilities, in order to increase the accessibility of health services for women with disabilities, this study aimed to address the considerable barriers they face when trying to access such services [50]. Hence, we need to pay more attention to the health conditions of women with disabilities, as well as to their access to appropriate healthcare services. Regarding limitations in the current study, although we conducted a suitable statistical method by using propensity score matching to create the homogenous characteristics of participants in two groups, methodological biases may still exist even after the analysis was conducted. Since women with disabilities were more prone to experience a lack of education, marriage, and health services [51,52,53], there may have been an imbalance in the variables, such as education level, marital status, and adults’ preventive care service between people with and without disabilities. Thus, these factors were not included in the matching process. Based on national data obtained from the NHIRD, this study did not include lifestyle risk factors or psychological risk factors; hence, these variables were not examined. However, this study conducted samples comprising a large number of individuals with disabilities that can represent the whole picture of people with disabilities in Taiwan.

## 5. Conclusions

In Taiwan, the proportion and probability of women with disabilities who underwent breast cancer screening were lower than the proportion and probability of those without disabilities. Health inequality in Taiwan corresponds with the WHO’s statements that disabled people around the world are at a significantly lower level of wellness than the non-disabled population. Therefore, continual efforts should be made to diminish health disparities between women with and without disabilities around the globe. Concerning the results of the current study, these findings could be a reference for policy development regarding breast cancer prevention, where public health nursing and female health specialists will be of high importance in encouraging women with disabilities to be given more opportunities to receive health education and healthcare services regarding breast cancer screening. Furthermore, caregivers or relatives could be trained to motivate those with disabilities to experience breast cancer screening at the suggested time to comply with the early detection recommendations for breast cancer. In some situations, it may be possible to establish an accreditation program to ensure that information, transportation facilities, and access to healthcare services are available to women with disabilities.

## Figures and Tables

**Figure 1 ijerph-19-05280-f001:**
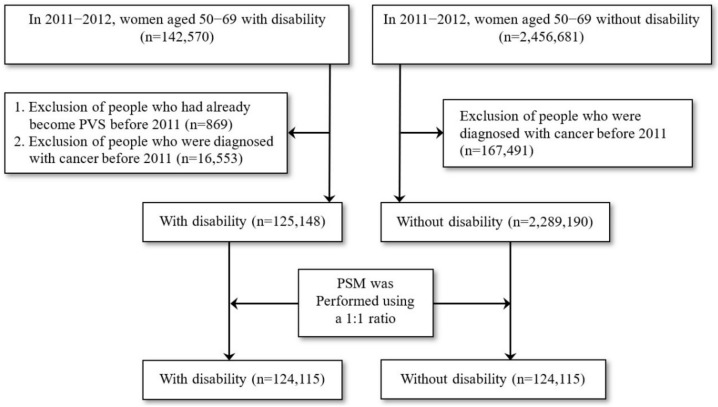
Study participants were selected and matched between women with and without disabilities.

**Table 1 ijerph-19-05280-t001:** The distribution in characteristics of women with disabilities and those without disabilities who meet the criteria for breast cancer screening after matching.

Variables			Women without Disabilities		Women with Disabilities		
	N	%	N	%	N	%	*p*-Value ^a^
**Total**	248,230	100.00	124,115	50.00	124,115	50.00	
**Age**							0.827
50–59 years	137,555	55.41	68,750	49.98	68,805	50.02	
60–69years	110,675	44.59	55,365	50.02	55,310	49.98	
**Monthly salary (NT$)**							0.976
Low-income households	6856	2.76	3397	49.55	3459	50.45	
≦17,280	14,642	5.90	7352	50.21	7290	49.79	
17,281–22,800	109,430	44.08	54,715	50.00	54,715	50.00	
22,801–28,800	50,570	20.37	25,285	50.00	25,285	50.00	
28,801–36,300	33,484	13.49	16,742	50.00	16,742	50.00	
≧36,301	33,248	13.39	16,624	50.00	16,624	50.00	
**Urbanization level**							1.000
1	50,986	20.54	25,493	50.00	25,493	50.00	
2	75,233	30.31	37,644	50.04	37,589	49.96	
3	37,336	15.04	18,668	50.00	18,668	50.00	
4	45,329	18.26	22,637	49.94	22,692	50.06	
5	8400	3.38	4200	50.00	4200	50.00	
6	16,311	6.57	8151	49.97	8160	50.03	
7	14,635	5.90	7322	50.03	7313	49.97	
**CCI ^b^**							0.990
0	105,748	42.60	52,843	49.97	52,905	50.03	
1	53,160	21.42	26,580	50.00	26,580	50.00	
2	38,196	15.39	19,098	50.00	19,098	50.00	
≧3	51,126	20.60	25,594	50.06	25,532	49.94	

^a^ Chi-square. ^b^ Charlson comorbidity index.

**Table 2 ijerph-19-05280-t002:** Comparison of the use of breast cancer screening between women with disabilities and those without disabilities.

Variables	Total		Women without Disabilities				Women with Disabilities				
			No Mammography		Mammography		No Mammography		Mammography		
	N	%	N	%	N	%	N	%	N	%	*p*-Value ^a^
**Total**	248,230	100.00	92,438	74.48	31,677	25.52	101,367	81.67	22,748	18.33	<0.001
**Disability type**											
Without disability	124,115	50.00	92,438	74.48	31,677	25.52					
Moving functional limitation	55,296	22.28					44,446	80.38	10,850	19.62	
Visual impairment	8201	3.30					6745	82.25	1456	17.75	
Hearing impairment	11,980	4.83					9025	75.33	2955	24.67	
Speech disorder	919	0.37					741	80.63	178	19.37	
Intellectual disability	4245	1.71					3676	86.60	569	13.40	
Multiple disabilities	9438	3.80					8449	89.52	989	10.48	
Dysfunction of primary organs	18,956	7.64					16,417	86.61	2539	13.39	
Facial impairment	214	0.09					149	69.63	65	30.37	
Dementia	1642	0.66					1529	93.12	113	6.88	
Congenital disorders	24	0.01					14	58.33	10	41.67	
Chronic mental health conditions	12,667	5.10					9761	77.06	2906	22.94	
Balance disorder	322	0.13					271	84.16	51	15.84	
Intractable epilepsy	151	0.06					96	63.58	55	36.42	
Rare diseases	42	0.02					33	78.57	9	21.43	
Other ^b^	18	0.01					15	83.33	3	16.67	
**Severity of disability**											
Without disability	124,115	50.00	92,438	74.48	31,677	25.52					
Mild	45,336	18.26					33,781	74.51	11,555	25.49	
Moderate	39,179	15.78					31,988	81.65	7191	18.35	
Severe	19,915	8.02					17,774	89.25	2141	10.75	
Very severe	19,685	7.93					17,824	90.55	1861	9.45	
**Age**											
50–59 years	137,555	55.41	44,142	64.21	24,608	35.79	50,326	73.14	18,479	26.86	<0.001
60–69 years	110,675	44.59	48,296	87.23	7069	12.77	51,041	92.28	4269	7.72	<0.001
**Monthly salary (NT$)**											
Low-income households	6856	2.76	2676	78.78	721	21.22	2978	86.09	481	13.91	<0.001
≦17,280	14,642	5.90	5656	76.93	1696	23.07	6216	85.27	1074	14.73	<0.001
17,281–22,800	109,430	44.08	41,645	76.11	13,070	23.89	45,427	83.02	9288	16.98	<0.001
22,801–28,800	50,570	20.37	18,791	74.32	6494	25.68	20,770	82.14	4515	17.86	<0.001
28,801–36,300	33,484	13.49	12,070	72.09	4672	27.91	13,295	79.41	3447	20.59	<0.001
≧36,301	33,248	13.39	11,600	69.78	5024	30.22	12,681	76.28	3943	23.72	<0.001
**Urbanization level**											
1	50,986	20.54	19,070	74.80	6423	25.20	20,967	82.25	4526	17.75	<0.001
2	75,233	30.31	27,837	73.95	9807	26.05	30,163	80.24	7426	19.76	<0.001
3	37,336	15.04	14,119	75.63	4549	24.37	15,662	83.90	3006	16.10	<0.001
4	45,329	18.26	16,927	74.78	5710	25.22	18,630	82.10	4062	17.90	<0.001
5	8400	3.38	3117	74.21	1083	25.79	3342	79.57	858	20.43	<0.001
6	16,311	6.57	5954	73.05	2197	26.95	6583	80.67	1577	19.33	<0.001
7	14,635	5.90	5414	73.94	1908	26.06	6020	82.32	1293	17.68	<0.001
**CCI ^c^**											
0	105,748	42.60	39,484	74.72	13,359	25.28	41,459	78.36	11,446	21.64	<0.001
1	53,160	21.42	19,293	72.58	7287	27.42	20,957	78.84	5623	21.16	<0.001
2	38,196	15.39	13,997	73.29	5101	26.71	15,879	83.14	3219	16.86	<0.001
≧3	51,126	20.60	19,664	76.83	5930	23.17	23,072	90.37	2460	9.63	<0.001
**Adults’ preventive care service**											
No	161,286	64.97	57,994	78.73	15,668	21.27	75,515	86.18	12,109	13.82	<0.001
Yes	86,944	35.03	34,444	68.27	16,009	31.73	25,852	70.84	10,639	29.16	<0.001

^a^ Chi-Square. ^b^ Including autism, chromosomal abnormalities, and congenital metabolic disorders. ^c^ Charlson comorbidity index.

**Table 3 ijerph-19-05280-t003:** Relevant factors affecting the use of breast cancer screening for women with disabilities and those without disabilities (conditional logistic regression analysis).

Variables	Model A				Model B				Model C			
	OR	95% CI		*p*-Value	OR	95% CI		*p*-Value	OR	95% CI		*p*-Value
**Disability status**												
No (reference)	1.00	–	–	–								
Yes	0.72	0.71	0.74	<0.001								
**Disability type**												
Without disability (reference)					1.00	–	–	–				
Moving functional limitation					0.73	0.68	0.78	<0.001				
Visual impairment					1.00	0.95	1.06	0.884				
Hearing impairment					0.71	0.59	0.87	0.001				
Speech disorder					0.77	0.74	0.79	<0.001				
Intellectual disability					0.45	0.41	0.50	<0.001				
Multiple disabilities					0.43	0.40	0.47	<0.001				
Dysfunction of primary organs					0.59	0.56	0.62	<0.001				
Facial impairment					0.97	0.68	1.39	0.857				
Dementia					0.34	0.28	0.43	<0.001				
Congenital disorders					1.95	0.61	6.26	0.263				
Chronic mental health conditions					0.82	0.78	0.86	<0.001				
Balance disorder					0.63	0.45	0.89	0.008				
Intractable epilepsy					1.33	0.90	1.97	0.154				
Rare diseases					0.65	0.28	1.53	0.322				
Other ^a^					0.38	0.09	1.56	0.181				
**Severity of disability**												
Without disability (reference)									1.00	–	–	–
Mild									1.00	0.97	1.03	0.906
Moderate									0.73	0.71	0.76	<0.001
Severe									0.42	0.40	0.45	<0.001
Very severe									0.39	0.37	0.41	<0.001
**Age**												
50–59 years (reference)	1.00	–	–	–	1.00	–	–	–	1.00	–	–	–
60–69 years	0.27	0.26	0.28	<0.001	0.27	0.26	0.28	<0.001	0.27	0.26	0.28	<0.001
**Monthly salary (NT$)**												
Low-income households (reference)	1.00	–	–	–	1.00	–	–	–	1.00	–	–	–
≦17,280	1.27	1.11	1.45	0.001	1.25	1.09	1.43	0.001	1.19	1.04	1.36	0.011
17,281–22,800	1.35	1.21	1.51	<0.001	1.32	1.18	1.48	<0.001	1.25	1.12	1.40	<0.001
22,801–28,800	1.32	1.17	1.48	<0.001	1.30	1.15	1.46	<0.001	1.23	1.10	1.39	0.001
28,801–36,300	1.33	1.18	1.50	<0.001	1.32	1.17	1.49	<0.001	1.26	1.11	1.42	<0.001
≧36,301	1.51	1.34	1.71	<0.001	1.48	1.31	1.67	<0.001	1.40	1.24	1.58	<0.001
**Urbanization level**												
1 (reference)	1.00	–	–	–	1.00	–	–	–	1.00	–	–	–
2	1.22	1.17	1.27	<0.001	1.22	1.17	1.27	<0.001	1.21	1.16	1.26	<0.001
3	1.07	1.02	1.13	0.006	1.08	1.03	1.13	0.003	1.08	1.03	1.13	0.004
4	1.32	1.26	1.39	<0.001	1.33	1.26	1.39	<0.001	1.31	1.25	1.37	<0.001
5	1.73	1.56	1.91	<0.001	1.72	1.56	1.91	<0.001	1.69	1.53	1.87	<0.001
6	1.77	1.64	1.91	<0.001	1.77	1.64	1.91	<0.001	1.74	1.61	1.87	<0.001
7	1.55	1.43	1.68	<0.001	1.56	1.44	1.69	<0.001	1.51	1.40	1.63	<0.001
**CCI ^b^**												
0 (reference)	1.00	–	–	–	1.00	–	–	–	1.00	–	–	–
1	1.36	1.30	1.42	<0.001	1.37	1.31	1.44	<0.001	1.35	1.29	1.41	<0.001
2	1.39	1.31	1.49	<0.001	1.45	1.36	1.55	<0.001	1.48	1.38	1.58	<0.001
≧3	1.17	1.09	1.25	<0.001	1.24	1.16	1.33	<0.001	1.29	1.21	1.38	<0.001
**Adults’ preventive care service**												
No (reference)	1.00	–	–	–	1.00	–	–	–	1.00	–	–	–
Yes	2.61	2.55	2.68	<0.001	2.56	2.49	2.62	<0.001	2.49	2.43	2.56	<0.001

^a^ Including autism, chromosomal abnormalities, and congenital metabolic disorders. ^b^ Charlson comorbidity index.

## Data Availability

Taiwan’s Ministry of Health and Welfare (MOHW) provides data through its Health and Welfare Data Science Center (https://www.mohw.gov.tw/mp-2.html (accessed on 5 August 2021)). The MOHW manages a database that is available to all researchers. In accordance with Taiwan’s Personal Information Protection Act, public access to the database is not permitted. Therefore, the authors were not able to make this data set publicly available.

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
