# Peer review of "Inequality in the Utilization of Breast Cancer Screening between Women with and without Disabilities in Taiwan: A Propensity-Score-Matched Nationwide Cohort Study"

_ijerph, 2022, doi:10.3390/ijerph19095280_

Round 1

Reviewer 1 Report

I have read with interest the paper from Inchai et al, focusing on breast cancer screening among women with disabilities in Taiwan. This is an interesting study, that offers a valuable perspective on mammography screening in the region.

Some comments as follows:

  • The article focused on organized mammography screening. What about opportunistic screening among women aged 50-69 in Taiwan. Is there any data?
  • In the study, the percentages of women undergoing mammography is around 20%. This finding is rather lower than reported for western countries, where screening adherence rates are stably above 50% in the same age range population. Perhaps, information campaigns should be made to enhance participation of women to screening mammography. This issues might be briefly added in the discussion.
  • In general, the discussion is too long. Lines 204-243 are dedicated only to the aspect of dementia. I wonder if that part can be shortened.
  • Furthermore,
  • The discussion lacks a paragraph on the study limitations. Although the statistical analysis were appropriate, the Propensity Score-Matched study design does have limitations. Besides, the data analysis referred to a relatively distant period (women with and without disabilities from 2004 to 2010 who received breast cancer screening during 2011 and 2012). This calls for precaution when interpreting data projection in the current years.
  • The sentence in lines 275-79 (Based on…) contains repetitions and should be rephrased.

ABSTRACT:

  • Since the main result is that the proportion of women with disabilities who received screening was significantly lower than women without disabilities (18.33 vs 25.52%), the corresponding p-value should be reported (the same should be done in the first sentence of the Discussion section).
  • The conclusion simply recapitulates the results of the study. A sentence addressing the possible measures that the government should take to ameliorate the screening participation of women with disabilities would be of benefit.

Figure 1: In the first two boxes the term ‘women’ should be added (…50-69 years women with…)

Tables: the acronym CCI should be defined in tables legend.

Author Response

Dear Reviewer:

We have revised the manuscript according to your comments. Please see the attached file. Thank you.

Reviewer 2 Report

This is paper on a very important topic such as the inequalities in breast cancer screening utilization in women with and without disabilities. Very interesting idea into the setting of public health.

However because of the great importance of this paper, some methodological points have to be addressed, firstly the presentation the propensity score analysis. According to some review, for example the paper “Reporting and Guidelines in Propensity Score Analysis: A Systematic Review of Cancer and Cancer Surgical Studies” (JNCI, 2017) a well structured reporting is essential to ensure the reproducibility, appropriateness and effectiveness of the results got by using propensity score (PS) analysis. To enhance the reporting of PS analysis and following Table 4 of the review I have cited, I suggest to the authors of the paper under review to add some specification about the following points:

- to clearly specify which variable were used for the propensity score matching and which for the outcome analysis

- in the methods, to explain which bias may be avoided by using the propensity score matching

- to specify which model was used to estimate propensity scores

- to specify which methods was applied to assess the comparability of baseline characteristics between the two matched groups

- to specify how missing values were treated in the context of propensity score and for the outcome analysis

- to describe the distribution of baseline characteristics for each group before and after propensity score matching

- report propensity score analysis estimates

- discuss whether imbalance of baseline characteristic still exist after matching

- address other points considered relevant to describe propensity score analysis

Other point:

the authors said that “excluded the study population who had breast cancer before 2011”:

It is mandatory to specify how the exclusion was performed, for example using a population-based cancer registry

Author Response

(The authors gave the same response as above.)

Reviewer 3 Report

I think the authors did an excellent piece of work. 

Only one comment regards adding at least one reference to one of the current hot topics regarding breast cancer screening. I am referring to mammograms analysis using CAD systems reliant on Artificial Intelligence. 

Author Response

(The authors gave the same response as above.)

Reviewer 4 Report

The article is appealing because it allows us to study whether a disability is a barrier to breast cancer screening.

The following changes are suggested

The reference in the introduction to the propensity score (lines 73 and 74) should be in material and methods.

The material and methods should explain the levels of urbanization in more detail. How they have been calculated, and, if necessary, provide a bibliographic reference.

It would be interesting to describe Taiwan's healthcare system briefly. This will allow a better understanding of the context of the article for readers who are not familiar with it.

The list of diseases in lines 127-133 corresponds to the Charlson Comorbidity Index. The list can be omitted as it makes the article difficult to read. It would suffice to say that the Charlson Index provided the reference. It could be indicated that the diseases that compose it are analyzed.

The abbreviations should be explained at the foot of the tables. The tables should be understandable without having to read the text of the article. This occurs in several tables in which the abbreviation CCI is used. 

Author Response

(The authors gave the same response as above.)
